# Trends and associated factors for Covid-19 hospitalisation and fatality risk in 2.3 million adults in England

T. Beaney [1,2 ✉], A. L. Neves[1], A. Alboksmaty[1,2], H. Ashrafian [1], K. Flott[1], A. Fowler[3], J. R. Benger[4], P. Aylin[1,2], S. Elkin[5], A. Darzi [1] & J. Clarke [1,6]

The Covid-19 mortality rate varies between countries and over time but the extent to which this is explained by the underlying risk in those infected is unclear. Using data on all adults in England with a positive Covid-19 test between 1st October 2020 and 30th April 2021 linked to clinical records, we examined trends and risk factors for hospital admission and mortality. Of 2,311,282 people included in the study, 164,046 (7.1%) were admitted and 53,156 (2.3%) died within 28 days of a positive Covid-19 test. We found significant variation in the case hospitalisation and mortality risk over time, which remained after accounting for the underlying risk of those infected. Older age groups, males, those resident in areas of greater socioeconomic deprivation, and those with obesity had higher odds of admission and death. People with severe mental illness and learning disability had the highest odds of admission and death. Our findings highlight both the role of external factors in Covid-19 admission and mortality risk and the need for more proactive care in the most vulnerable groups.

[1] Patient Safety Translational Research Centre, Institute of Global Health Innovation, Imperial College London, London SW7 2AZ, UK. [2] Department of Primary Care and Public Health, Imperial College London, London W6 8RP, UK. [3] NHS England and Improvement, London SE1 6LH, UK. [4] NHS Digital, 7-8 Wellington Place, Leeds, West Yorkshire LS1 4AP, UK. [5] National Heart and Lung Institute, Imperial College London, London SW7 2AZ, UK. [6] Centre for Mathematics of Precision Healthcare, Department of Mathematics, Imperial College London, London SW7 2AZ, UK. ✉email: thomas.beaney@imperial.ac.uk

The Covid-19 case fatality ratio (CFR) varies widely between countries[1] and definitions of mortality differ across the world, making comparisons challenging[2]. In England, the most widely reported measure is mortality within 28 days of a positive test[3]. Up to 21 September 2021, 539,921 hospital admissions and 118,846 deaths have occurred in England, out of a total of 6,398,633 cases, giving a crude case hospitalisation ratio (CHR) of 8.4% and a CFR of 1.9%[4]. Previous epidemiological studies have shown variation in the CFR over time[1,5], but without individual level data, it is unclear to what extent this variation is accounted for by differences in the risk of those infected.

Many risk factors for death from Covid-19 have been characterised, such as increased age, male gender, and obesity[6]. Several long-term conditions are strongly linked to a higher mortality risk; in England, this led to the early adoption of a 'clinically extremely vulnerable' (CEV) status for those deemed to be at highest risk, subsequently advised to isolate to reduce transmission[7]. Previous studies have focussed on the first wave of the pandemic in the first half of 2020, which may not be representative of subsequent pandemic waves, particularly given advances in the management of Covid-19 patients and the emergence of new variants[8]. Furthermore, to our knowledge, no study to date has used data with national coverage, including all laboratory-confirmed Covid-19 test results linked to electronic health record (EHR) data.

The main aim of this paper is to describe the changing trends in the Covid-19 case hospitalisation risk (CHR) and case fatality risk (CFR) in England, during the 'second wave' of the pandemic (i.e., from 1st October 2020 to 30th April 2021). The secondary aims are to identify patient characteristics associated with hospitalisation and mortality risk; and to evaluate whether residual unexplained variation in the CHR and CFR remains after accounting for differences in the underlying risk factors of those infected.

## Results

From 1st October 2020 to 30th April 2021, data were available for 2,433,768 individuals with a positive Covid-19 test result in England. Data for 34,317 (1.4%) participants with a positive test result could not be linked to either primary or secondary care records and were excluded. Care home residents accounted for 3.7% of the total (n = 88,169) and were excluded from further analyses, resulting in a total population of 2,311,282.

Characteristics of the study population are provided in Table 1. The mean (SD) age of participants was 44.3 (17.1) years, with 43.6% under 40 years. The majority were female (54.3%) and of White ethnicity (72.8%). There were relatively higher proportions from more deprived deciles of IMD, with 56.7% in the bottom five deciles. Similar proportions of subjects with a healthy weight (28.4%), overweight (28.1%) or obese (26.1%) were observed, and only 3.4% were underweight. 16.3% were current smokers and 8.3% were designated as CEV. Chronic respiratory disease (21.2%), hypertension (15.0%) and diabetes (8.6%) were the three most prevalent chronic conditions in the population.

**Case hospitalisation and fatality risk over time**. Of the study population, 164,046 people were admitted to hospital at least once within 28 days of a positive test, giving a crude CHR of 7.1% over the seven-month period. 53,156 deaths occurred within 28 days of a positive test, giving a crude CFR of 2.3%. Of these, 49,172 (92.5%) had Covid-19 as a cause of death on the death certificate. There were significant differences over time in both the CHR and CFR (Supplementary Fig. 1). The age distribution of people with a positive test varied over time, with the highest proportions of all infection in people aged 60 years and

above infected in November 2020 and January 2021 (Supplementary Table 1). Within all age groups, a similar pattern of change in the CHR and CFR over time was seen, with risk peaking in December 2020–February 2021 (Supplementary Tables 2 and 3, respectively, and Supplementary Fig. 2).

**Factors associated with 28-day mortality and hospitalisation risk**. Multiple imputation was used to impute missing data for 381,283 people. Multivariable logistic regression models were constructed for each outcome adjusting for all patient level covariates (model 2). Calibration plots indicated adequate calibration (Supplementary Figs. 3 and 4). Results for hospital admissions and mortality are presented in Figs. 1 and 2 (also Supplementary Tables 4 and 5). Males had 41% higher adjusted odds of admission (95% CI: 1.39–1.42) and 62% higher adjusted odds of mortality (95% CI: 1.58–1.65) compared to females. People of all four non-White ethnicities had higher odds of admission, and those of Asian and Black ethnicities also had higher odds of mortality compared to those of White ethnicity. People living in less deprived areas had lower odds of both admission and mortality compared to those in the most deprived areas. Compared to people of a healthy weight, those underweight had 10% higher odds of admission (95% CI: 1.05–1.14) and 99% higher odds of death (95% CI: 1.87–2.11). People who were overweight had a 24% increase in odds of admission (95% CI: 1.22–1.26) but 20% lower odds of death (95% CI: 0.77-0.82); those who were obese had 93% higher odds of admission (1.90–1.97) and 4% increased odds of death (95% CI: 1.01–1.07). Current smokers had lower odds of admission compared to non-smokers but an increase in the odds of death after adjustment.

All chronic conditions included were strongly associated with an increase in odds of admission and death, except for dementia, which was associated with 6% lower odds of admission. People identified as CEV had 85% higher odds of being admitted to hospital (95% CI: 1.83–1.88) but 12% lower odds of death (95% CI: 0.86–0.90) after full adjustment. In a sub-analysis adjusting CEV status for age, time (and their interaction), sex, ethnicity, and deprivation only, odds of admission were significantly higher (aOR 2.62, 95% CI: 2.58–2.65) as were odds of death (aOR 1.52, 95% CI: 1.49–1.55).

A sensitivity analysis of the 1,929,999 complete cases showed similar estimates to the fully adjusted model (Supplementary Tables 6 and 7).

**CHR and CFR over time**. A significant association remained with time for both CHR and CFR models after adjusting for all patient covariates (p < 0.0001 in each model from likelihood ratio tests). The predicted CHR and CFR from the fully adjusted models are plotted for the whole population (Supplementary Fig. 5) and by age category in Fig. 3, showing that a significant time-varying relationship remained after adjustment. The relative change in predicted CHR and CFR from the baseline predicted risk in the first full week of October is shown in Fig. 4 (and Supplementary Figs. 6 and 7). The CFR increased across all age groups, peaking between late December 2020 to early February 2021in different age groups before declining towards April. A smaller relative increase in hospitalisation risk was seen across age groups. In most age groups, CHR peaked in January, except in the 18–39 age group, which continued to increase throughout the study period. After adjustment, the trends in absolute mortality and hospitalisation risk in each age group were similar to those in the unadjusted analyses (Fig. 4 and Supplementary Fig. 2) indicating that the distributions of risk factors of those infected within age groups did not change significantly over time.

**Table 1 Characteristics of the study population with hospital admissions and deaths within 28 days ($N = 2,311,282$).**

| | Total | | Hospital admissions | | Deaths | |
|---|---|---|---|---|---|---|
| | Number | Percentage | Number | Percentage | Number | Percentage |
| *Age category (years) and CEV status* | | | | | | |
| Mean (SD) | 44.3 (17.1) years | | | | | |
| 18–39 | 1,007,474 | 43.6% | 19,834 | 2.0% | 429 | 0.1% |
| 40–49 | 442,337 | 19.1% | 18,897 | 4.3% | 989 | 0.7% |
| 50–59 | 434,690 | 18.8% | 30,138 | 6.9% | 3,054 | 0.7% |
| 60–69 | 229,209 | 9.9% | 30,070 | 13.1% | 7,009 | 3.1% |
| 70–79 | 112,379 | 4.9% | 31,436 | 28.0% | 14,068 | 12.5% |
| 80 or older | 85,193 | 3.7% | 33,671 | 39.5% | 27,607 | 32.4% |
| *Sex* | | | | | | |
| Female | 1,255,364 | 54.3% | 72,126 | 5.7% | 21,253 | 1.7% |
| Male | 1,010,045 | 43.7% | 86,135 | 8.5% | 29,574 | 2.9% |
| Missing | 45,873 | 2.0% | 5,785 | 12.6% | 2,329 | 5.1% |
| *Ethnicity* | | | | | | |
| White | 1,681,477 | 72.8% | 119,999 | 7.1% | 42,753 | 2.5% |
| Asian/Asian British | 304,685 | 13.2% | 21,900 | 7.2% | 4,539 | 1.5% |
| Black/African/Caribbean/Black British | 87,974 | 3.8% | 7,880 | 9.0% | 1,505 | 1.7% |
| Mixed/multiple ethnic groups | 38,397 | 1.7% | 2,236 | 5.8% | 392 | 1.0% |
| Other ethnic group | 58,789 | 2.5% | 4,388 | 7.5% | 616 | 1.0% |
| Missing | 139,960 | 6.1% | 7,643 | 5.5% | 3,351 | 2.4% |
| *Index of Multiple Deprivation (IMD) decile* | | | | | | |
| 1 (most deprived) | 277,814 | 12.0% | 23,009 | 8.3% | 6,734 | 2.4% |
| 2 | 282,141 | 12.2% | 22,238 | 7.9% | 6,510 | 2.3% |
| 3 | 271,120 | 11.7% | 20,001 | 7.4% | 5,985 | 2.2% |
| 4 | 248,041 | 10.7% | 17,838 | 7.2% | 5,689 | 2.3% |
| 5 | 231,591 | 10.0% | 16,376 | 7.1% | 5,220 | 2.3% |
| 6 | 218,370 | 9.4% | 14,774 | 6.8% | 5,104 | 2.3% |
| 7 | 208,612 | 9.0% | 13,647 | 6.5% | 4,831 | 2.3% |
| 8 | 206,003 | 8.9% | 13,316 | 6.5% | 4,719 | 2.3% |
| 9 | 194,561 | 8.4% | 12,273 | 6.3% | 4,440 | 2.3% |
| 10 (least deprived) | 172,508 | 7.5% | 10,542 | 6.1% | 3,917 | 2.3% |
| Missing | 521 | <0.1% | 32 | 6.1% | 7 | 1.3% |
| *Body mass index* | | | | | | |
| Underweight | 78,684 | 3.4% | 3,149 | 4.0% | 2,063 | 2.6% |
| Healthy weight | 655,582 | 28.4% | 30,448 | 4.6% | 13,967 | 2.1% |
| Overweight | 649,641 | 28.1% | 48,161 | 7.4% | 15,652 | 2.4% |
| Obese | 603,303 | 26.1% | 67,638 | 11.2% | 17,199 | 2.9% |
| Missing | 324,072 | 14.0% | 14,650 | 4.5% | 4,275 | 1.3% |
| *Smoking status* | | | | | | |
| Never smoker | 1,363,771 | 59.0% | 82,668 | 6.1% | 21,963 | 1.6% |
| Ex-smoker | 475,558 | 20.6% | 53,186 | 11.2% | 21,757 | 4.6% |
| Current smoker | 376,057 | 16.3% | 21,260 | 5.7% | 7,005 | 1.9% |
| Missing | 95,896 | 4.1% | 6,932 | 7.2% | 2,431 | 2.5% |
| *Clinically Extremely Vulnerable* | 192,531 | 8.3% | 48,679 | 25.3% | 19,294 | 10.0% |
| *Co-morbidities* | | | | | | |
| Hypertension | 346,145 | 15.0% | 69,202 | 20.0% | 31,710 | 9.2% |
| Chronic cardiac disease | 126,133 | 5.5% | 37,775 | 29.9% | 22,878 | 18.1% |
| Chronic kidney disease | 14,492 | 0.6% | 5,485 | 37.8% | 3,553 | 24.5% |
| Chronic respiratory disease | 489,341 | 21.2% | 53,099 | 10.9% | 19,960 | 4.1% |
| Dementia | 13,552 | 0.6% | 5,111 | 37.7% | 4,800 | 35.4% |
| Diabetes | 199,495 | 8.6% | 44,280 | 22.2% | 18,293 | 9.2% |
| Chronic neurological disease (including epilepsy) | 64,274 | 2.8% | 10,765 | 16.7% | 4829 | 7.5% |
| Learning disability | 11,627 | 0.5% | 1,775 | 15.3% | 485 | 4.2% |
| Malignancy or immunosuppression | 178,648 | 7.7% | 30,014 | 16.8% | 15,691 | 8.8% |
| Severe mental illness | 39,513 | 1.7% | 7,038 | 17.8% | 3,684 | 9.3% |
| Peripheral vascular disease | 14,453 | 0.6% | 5,266 | 36.4% | 3,664 | 25.4% |
| Stroke or TIA | 44,239 | 1.9% | 13,880 | 31.4% | 8,712 | 19.7% |
| *Total* | 2,311,282 | | 164,046 | | 53,156 | |

## Discussion

In this retrospective cohort study including all adults in England with a positive Covid-19 test result, there was significant variation in the 28-day CHR and CFR by age group and over time, which remained after accounting for individual risk. Demographics and chronic conditions were strongly associated with hospitalisation and death.

**Variation in CHR and CFR over time**. Across the whole study population, CHR and CFR varied over time from 1st October

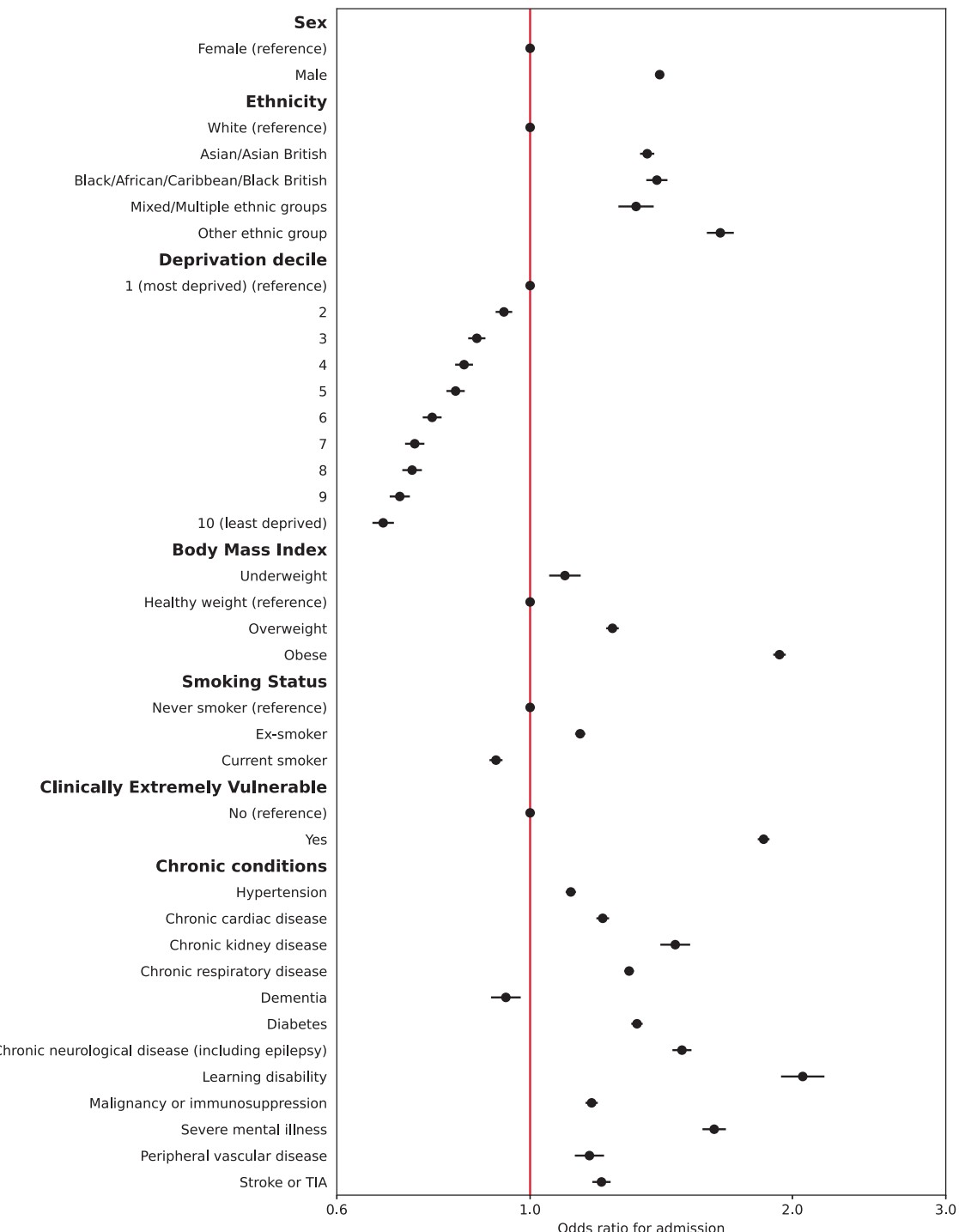

**Fig. 1 Adjusted odds ratios for emergency hospital admission within 28 days of positive Covid-19 test.** Circles represent odds ratios and error bars represent the 95% confidence intervals for patient-level predictors from multivariable mixed effects logistic regression models ($N = 2,311,282$).

2020 to 30th April 2021. This was partially explained by the changing age distributions of those infected, but significant variation remained after adjustment. Within age groups, absolute differences in the CHR and CFR over time were greatest in older age groups, reflecting higher baseline risk, but the relative risk varied significantly across all groups. Historically, there is a strong seasonal component to mortality in England, with figures indicating 16.8% higher mortality in winter months compared to summer months[9]. An increased incidence of respiratory diseases, including influenza, are one of the main drivers of increased winter mortality, and the 28-day mortality metric used in this study includes deaths from non-Covid-19 causes. However, with influenza rates at lower levels than previous years, it is unlikely the variation in CFR over time can be explained by the incidence of other infectious diseases alone[10].

Strain on the health system may also contribute to the patterns seen, with Covid-19 bed occupancy and critical care occupancy in England peaking in January 2021, associated with a lower proportion of patients seen in Accident & Emergency departments within 4 hours than in November 2020 and February 2021[4,11]. Larger relative increases were seen in the CFR compared to the CHR, which may indicate a health system reaching full

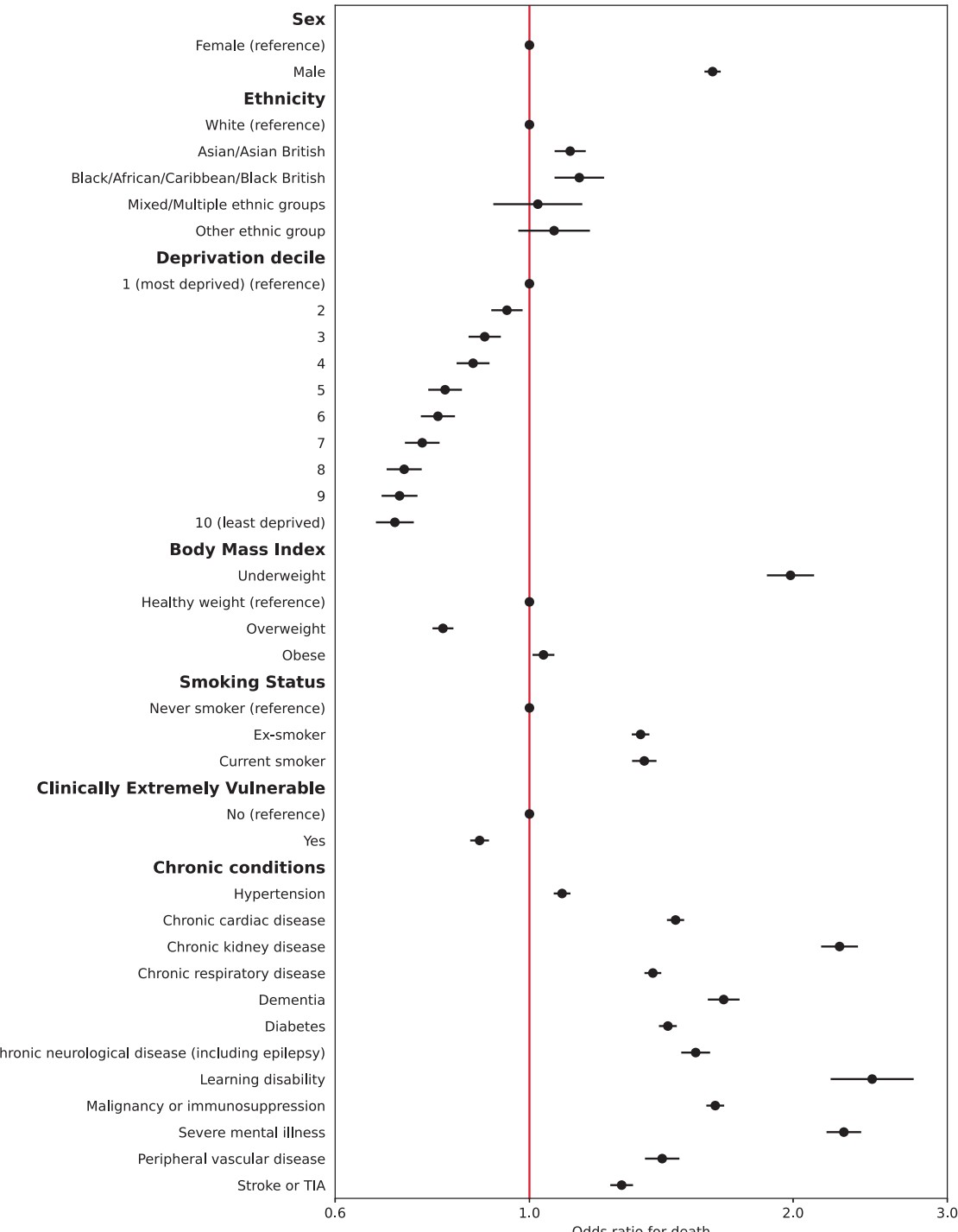

**Fig. 2 Adjusted odds ratios for death within 28 days of positive Covid-19 test.** Circles represent odds ratios and error bars represent the 95% confidence intervals for patient-level predictors from multivariable mixed effects logistic regression models ($N = 2,311,282$).

capacity and struggling to meet demand. A previous UK study of patients admitted to hospital with Covid-19 found a fall in mortality from March to July 2020, a time over which bed occupancy fell and evidence for new treatments, such as dexamethasone, became available, with similar findings from a US cohort between March and September 2020[12,13]. Changes to care delivery at an organisational level may also have an impact, with triage models for Covid-19 patients on the national 111 urgent care service varying between services and over time[14]. The Alpha variant became the dominant Covid-19 strain in England in December 2020, and has been associated with a 64% increase

in 28-day mortality compared to prior variants, which may explain part of the rise in the CHR and CFR[15].

Declines in the CHR and CFR from January 2021 onwards are likely to be explained at least partially by the development of immunity, both through natural infection and by the vaccination programme, which was implemented from 8th December 2020 in England for the highest risk cohorts[16]. By February 2021, over 80% of over 80s had been vaccinated in most regions of the UK, with similar vaccine coverage in the 70–79 year age group by mid-February and in the 60–69 year age group by mid-March (Supplementary Figs. 8–10)[17]. However, our study population

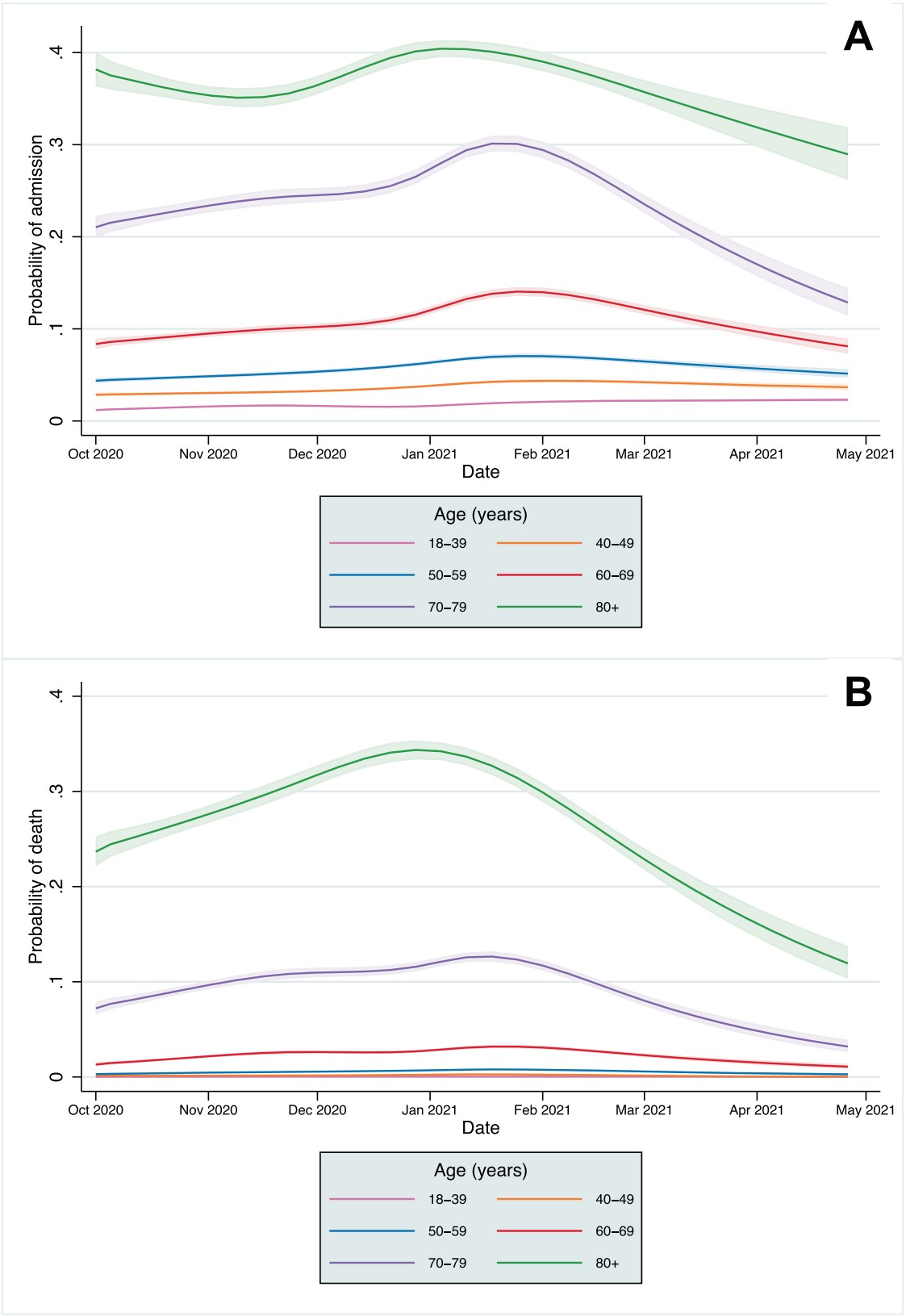

**Fig. 3 28-day case hospitalisation risk and fatality risk over time in people with Covid-19. A** 28-day case hospitalisation risk. **B** 28-day case fatality risk. Lines represent the probability estimate for each age group and shaded areas represent 95% confidence intervals from mixed effects logistic regression models, adjusted for patient-level covariates, at mean levels of each covariate (*N* = 2,311,282).

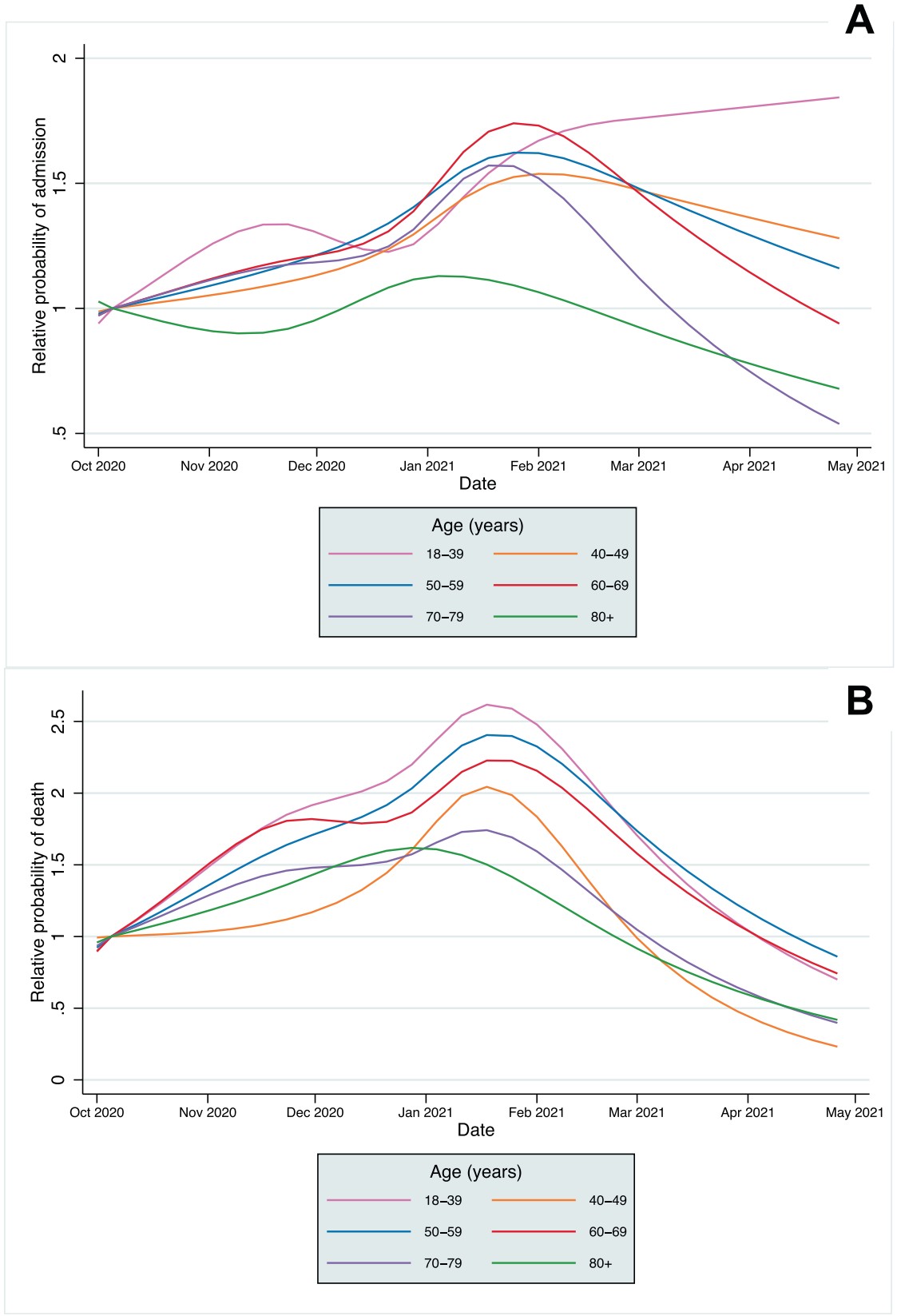

**Fig. 4 Relative change in 28-day case hospitalisation risk and fatality risk over time in people with Covid-19. A** 28-day case hospitalisation. **B** 28-day case fatality risk. Lines represent the probability estimate for each age group from mixed effects logistic regression models, adjusted for patient-level covariates, at mean levels of each covariate (*N* = 2,311,282). Y-axis is the probability relative to the first full week of October; note different scales for admissions and mortality.

includes people with a positive Covid-19 test, who are more likely to be unvaccinated than the general population; population vaccine coverage is, therefore, unlikely to be representative of our study population and estimates could not be incorporated robustly into our modelling. Declines in CFR and CHR are most marked in older age groups, who were the first groups eligible for vaccination. However, declines in mortality are seen across all age groups, including the 18–39 year group, many of whom would not have been eligible for vaccination, suggesting vaccination does not fully account for the declines observed. Availability of new treatments may also explain the falls in mortality, with the RECOVERY trial demonstrating the benefit of tocilizumab published in February 2021, but is unlikely to explain the fall in admissions[8,18].

**Factors associated with hospitalisation and mortality.** The findings of a higher risk of mortality in males, people of Asian and Black ethnic backgrounds, and those living in more deprived areas are consistent with a previous UK cohort and confirmed in our study, including an increased risk of admission[6]. People who were underweight were more likely to be admitted and had significantly higher risk of death, which might be partly accounted for by unmeasured associated conditions, such as frailty. People who were overweight and obese had higher risk of admission than those of a healthy weight, but mortality risk was lower in those overweight, which may indicate higher perceived risk amongst clinicians and a lower threshold for admission.

People identified as CEV were significantly more likely to be admitted but were found to have significantly lower mortality, after adjusting for other risk factors including co-morbidities. However, in partially adjusted models not including BMI, smoking, or clinical co-morbidities, those identified as CEV had significantly higher odds of death. Taken together, these findings indicate a lower threshold for clinical assessment and/or admission and escalation in CEV patients with a protective effect on mortality. All twelve included clinical co-morbidities were associated with significant increases in the odds of mortality and admission. Severe mental illness and learning disability had the strongest associations with mortality and admission, highlighting a need for more proactive care in these groups and more research into the reasons for mortality differences[19]. Those with dementia had significantly increased odds of mortality but were less likely to be admitted, suggesting they are more likely to receive care at home, although the cohort did not include those living in care homes and so will not represent the full population of those with dementia.

The emergence of the Delta and Omicron variants have shown the potential of Covid-19 to vary in both transmissibility and pathogenicity over time. In England, December 2021–January 2022 saw the highest case numbers but without the resulting number of hospitalisations and deaths associated with earlier variants and before widespread vaccination[4]. Despite the emergence of new variants, the findings of our study are relevant in highlighting that the risk of mortality was independent of an extensive panel of clinical and demographic factors in the winter of 2020/21, pointing to the role of wider strain on the health system as an important feature in outcomes in people with Covid-19. While the Omicron variant has contributed to an increase in hospitalisations and emergency department presentations in England and elsewhere, its impact on staff absence has been particularly marked. At the peak of the Omicron wave in early January 2022, almost 50,000 NHS staff were absent due to Covid-19, almost a five-fold increase from the end of November 2021[20–22]. Ensuring health systems possess the resilience to weather the dual shocks of an increased demand for care and

decreased capacity to provide it, without adversely affecting the quality and safety of healthcare, is an ongoing area of concern.

**Strengths and limitations.** A strength of this study is the inclusion of routine national laboratory data for positive Covid-19 test results in adults in England with only 1.5% unable to be linked to EHR data, and as a result, has lower risk of sampling bias[23]. To our knowledge, this is the largest such study including individual level data at a national level. Previous studies in England on predictors of mortality are reported on a smaller cohort of patients with 40% national coverage[6]. The use of multiple imputation assumes that data are missing at random, and we cannot rule out non-random missing patterns, particularly for data on ethnicity and deprivation, where more marginalised groups are less likely to be registered in the primary care record. However, sensitivity analyses showed inferences were similar between the complete case analysis and imputed results, suggesting limited impact of the missing data on model estimates. Associations with risk factors may also be confounded by differential uptake of vaccinations among risk groups; for example, if those with co-morbidities or defined as CEV were more likely to be vaccinated, the odds ratios for hospitalisation and death may be under-estimated.

Data represented here include only those who died within 28 days of a positive test result, in line with estimates reported by PHE. Deaths mentioning Covid-19 on a death certificate are an alternative metric used widely in many countries as recommended by the World Health Organisation[24] and have tended to give a larger estimate of deaths in England, due to those attributable to Covid-19 after 28 days[4]. Over 90% of deaths within 28 days in our study also had Covid-19 as a cause of death on the death certificate, but we did not have corresponding data for those cases recorded on a death certificate without a positive Covid-19 test. The associations found in our study might be different if using deaths recorded on death certificates, rather than deaths within 28 days of a positive Covid-19 test, particularly if there were changes to death certification practices over time.

Through use of linked EHR data, we were able to incorporate detailed medical factors for the study cohort. However, we were unable to explore the relationship with external factors such as Covid-19 variants. Geographical and time-varying system factors, such as proximity to a hospital and hospital capacity are likely to impact on a person's health-seeking behaviour. Our study included people living in the community and given patients in England may attend any hospital, and the size of hospital markets vary considerably across the country, we could not reliably model the impact of nearby hospital bed availability at an individual level. However, our modelling showed only minimal residual variation accounted for by CCG level clustering (intraclass correlation coefficient <1%), suggesting these additional factors would have minimal impact on the findings. Access to testing may also impact the probability of having a positive test. Positivity rates in England peaked on 31st December 2020 at 18.3% and fell to 1.7% by 1st April 2021[4], but the extent to which this reflects increased incidence or a lack of test availability is uncertain. It is possible that if testing were limited during the peak in cases in December 2020–January 2021, those with more symptomatic disease may have been more likely to receive a test, compared to those who were asymptomatic or with mild symptoms. This may in turn lead to an apparent increase in risk of mortality due to changes in the severity of illness of those testing positive, rather than the severity of disease within the community as a whole. Furthermore, access to testing may be driven by sociodemographic factors, and the finding of lower

hospitalisation and mortality risk in less deprived areas could reflect better availability of testing. Exploring mortality risk in patients admitted to hospital or to intensive care units and whether this changed over time was outside the scope of the current study but is an area for further research.

The risk of hospitalisation and death from Covid-19 varied significantly over time from October 2020 to April 2021 in all age groups, independent of the underlying risk in those infected. Time-varying risks should be considered by researchers and policymakers in assessing the risks of hospitalisation and mortality from Covid-19. People with severe mental illness and learning disability were amongst those with the highest odds of both admission and mortality, indicating the need for proactive care in these groups.

## Methods

The work was conducted as part of a wider service evaluation, approved by Imperial College Healthcare Trust on December 3rd 2020. Data access was approved by the Independent Group Advising on the Release of Data (IGARD; DARS-NIC-421524-R0Y3P) on April 15th 2021.

**Study design and population**. We conducted a retrospective cohort study including all adults (≥18 years) resident in England with a positive Covid-19 test result (polymerase chain reaction or lateral flow tests) from 1st October 2020 to 30th April 2021, excluding people resident in care homes. Study participants were followed-up for 28 days from the date of a first positive test. The two primary outcomes were (i) one or more emergency hospital admissions and (ii) death from any cause, each within 28 days from the date of the positive test.

**Data sources and data processing**. Several datasets were linked for this study and provided by NHS Digital as part of an evaluation of the NHS England Covid Oximetry @home programme[25]. Covid-19 testing data was sourced from the Public Health England (PHE) Second Generation Surveillance System[26], the national laboratory reporting system for positive Covid-19 tests, covering the period from 1st October 2020 to 30th April 2021. Primary care data came from the General Practice Extraction Service (GPES) Data for Pandemic Planning and Research (GDPPR)[27]. CEV status was linked to GDPPR from the Shielded Patient List (see Supplementary Methods)[28]. Data on hospital admissions came from Hospital Episode Statistics (HES) data set up to 31st May 2021, linked to Office for National Statistics (ONS) data on death registrations up to 5th July 2021. Datasets were linked using a de-identified NHS patient ID. Participants who could not be linked from testing data to at least one of GDPPR or HES were excluded.

Patient demographics were derived from GDPPR, or where missing, from HES. Lower layer super output area (LSOA) of residence was linked to indices of relative deprivation using deciles of Index of Multiple Deprivation (IMD) 2019[29]. Residence in a care home, CEV status, body mass index (BMI), and smoking status were derived from GDPPR only. BMI was categorised as underweight (<18.5 kg/m²), healthy weight (18.5–24.9 kg/m²), overweight (25.0–29.9 kg/m²) and obese (≥30.0 kg/m²). Chronic conditions were extracted from GDPPR based on Systematised Nomenclature of Medicine Clinical Terms (SNOMED-CT) codes pertaining to relevant diagnosis code clusters. Only codes recorded prior to the date of a positive Covid-19 test were included, to exclude any diagnoses following Covid-19 infection. Where the latest code indicated resolution of a condition, the diagnosis was excluded for that individual. Further details on data curation are given in the Supplementary Methods.

**Statistical analysis**. Patients were followed from date of first positive Covid-19 test to emergency hospital admission or death within 28 days. Mixed effects logistic regression was conducted for each outcome, with a two-level hierarchical model incorporating Clinical Commissioning Group (CCG, of which there are 106 in England) of residence as a random intercept. Time, represented by the week of Covid-19 test, was modelled as a restricted cubic spline with five knots placed at equally spaced percentiles[30]. Two models were run for each outcome:

1. Model 1: incorporating age category and time splines along with their interaction.
2. Model 2: incorporating age category and time splines along with their interaction and including all additional patient level covariates: sex, ethnicity, IMD decile, BMI category, CEV status, smoking status, and presence of chronic conditions.

For model 2, multiple imputation using chained equations was used to impute missing values of covariates, under the assumption that values were missing at random. All variables included in the analysis model were included in the imputation model[31]. Fifteen imputations were created, with a burn-in of 10 iterations which gave adequate precision and convergence, respectively (Supplementary Methods). A sensitivity analysis was performed using complete

cases only. Calibration was assessed using plots of predicted against observed probabilities for each decile of predicted probability.

For each outcome, the predicted probability of the outcome was computed within each age group and study week stratum to calculate age- and time-specific case hospitalisation risk (CHR) and case fatality risk (CFR). These were calculated using the fixed portion of the model (assuming zero random effects). The relative changes in the CHR and CFR over time were calculated as the predicted probability in each week relative to the week of 5th–11th October 2020 in each age group. In adjusted models (model 2), other model covariates were set to the population mean (or proportion for categorical variables) within each age group. For CEV status, an additional sub-analysis was conducted adjusting only for the age category and time splines (and their interaction), sex, ethnicity, and IMD decile. Further details of the statistical methods are given in Supplementary Methods.

Analyses were conducted in the Big Data and Analytics Unit Secure Environment, Imperial College, using Python version 3.9.5, Pandas version 1.2.3, and Stata version 17.0 (StataCorp).

## Data availability

The patient level data used in this study are not publicly available but are available to applicants meeting certain criteria through application of a Data Access Request Service (DARS) and approval from the Independent Group Advising on the Release of Data. Further information is given below: https://digital.nhs.uk/about-nhs-digital/corporate-information-and-documents/independent-group-advising-on-the-release-of-data.

## Code availability

The SNOMED terms used in defining chronic conditions are available in our GitHub repository: https://github.com/tbeaney/Imperial-COh-evaluation. Further analysis codes are available on request to the corresponding author.

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

## Acknowledgements

The authors would like to thank Gianluca Fontana, Saira Ghafur, Melanie Leis, and Mahsa Mazidi for their input and support. Data management was provided by the Big Data and Analytical Unit (BDAU) at the Institute of Global Health Innovation (IGHI), Imperial College London. The authors would like to thank the other evaluation teams collaborating on the CO@h programme: the National Institute for Health Research (NIHR) BRACE and RSET team, the Nuffield Trust and the Improvement Analytics Unit, a partnership between The Health Foundation and NHS England. The authors also extend their thanks to NHS Digital for data support and advice throughout the project. The authors acknowledge support from NHS England and NHS Improvement, the National Institute for Health Research (NIHR) Imperial Biomedical Research Council (BRC) and the NIHR Imperial Patient Safety Translational Research Centre. T.B. acknowledges support from the NIHR Applied Research Collaboration Northwest London. J.C. acknowledges support from the Wellcome Trust (215938/Z/19/Z). The study funders did not play a role in study design; in the collection, analysis, and interpretation of data; in the writing of the report; and in the decision to submit the article for publication. In addition, researchers were independent from funders, and all authors had full access to all of the data included in this study and can take responsibility for the integrity of the data and the accuracy of the data analysis.

## Author contributions

T.B., A.L.N., A.A., H.A., K.F., A.F., J.R.B., P.A., S.E., A.D., and J.C. were involved in the conceptualisation of the study and interpretation of the findings. T.B., A.L.N., K.F., J.R.B., P.A., S.E., A.D., and J.C. developed the study design and methodology. T.B. and J.C. conducted the data management and statistical analysis, supported by A.L.N. T.B. wrote the first draft of the manuscript, and all authors were involved in reviewing and editing the manuscript. All authors approved the final manuscript and decision for publication.

## Competing interests

H.A. is Chief Scientific Officer, Preemptive Medicine and Health Security at Flagship Pioneering. S.E. has received fees for an educational lecture sponsored by Astra Zeneca and is co-clinical director for the NHS England and Improvement London Respiratory Network. J.C. has received fees from Philips UK Limited for consultancy services outside of the submitted work. All other authors declare no competing interests.
