## [Peer Review File · Nature Communications]

Trends and associated factors for Covid-19 hospitalisation and fatality risk in 2.3 million adults in EnglandREVIEWER COMMENTS

Reviewer #1 (Remarks to the Author):

Summary

The manuscript from Beaney et al. is a well written report describing patient-level factors that are associated with hospitalization and death associated with COVID-19 over a 6-month period between October 2020 through April 2021. The major conclusions are relatively consistent with what has been previously published with this large cohort. While the findings of the paper are largely confirmatory re: existing hypotheses or published results, though in a novel data set, if the major comments below are able to be addressed, this would significantly improve the overall impact of the paper and assist in overcoming the limitations highlighted by the authors in the discussion.

Major Comments

1. The biggest effect modifier for COVID related hospitalization is likely to be the rollout of the vaccination program that overlaps with the observational period, yet this remains relatively unaccounted for in the analysis. The authors describe this appropriately within the discussion as a limitation affecting the interpretation of their results. I think given the known protective benefits from vaccines it's hard to disentangle the periodic time effects vs. introduction of significant new variables that can impact survival/hospitalization. Given there is EHR data on all these patients, is information on vaccination status not available to be added? In the absence of the data maybe there are statistical ways to handle the potential modifying effect. Could analyses consider weighted variable approaches that may help account for likelihood to receive a vaccine – the authors state that the older individuals were the most likely to receive a vaccine such that perhaps age has a differential weight in models according to the time period?
2. The other major limitation of course is that the authors consider all-cause mortality during the 28-day period of admission. While the authors discuss this limitation, is there no data available on the cause of death? Understanding that post-covid sequelae may result in varying contributions to death, it would still be important data for a reader to understand major cited causes of death, especially among those with significant comorbidity to understand potential differences in risk.
3. Is there a way to do a time-to-event analysis based on the data at hand to understand whether the comorbidities or other factors are associated with greater patient decompensation once admitted? I am envisioning a hazards model where the time-to-event model could be adjusted for the period (month or even week) the patients were admitted with COVID. This may help further disentangle some of the time effects from evolving secular trends (i.e. changes in treatment; availability of vaccines; etc.) or major circulating variant (emergence of B.1.1.7).
4. More discussion of the clinical relevance of these findings today would be helpful given that we have since moved on from B.1.1.7, experienced/ing delta, and now omicron has swept the world with another variant emerging – all of which seem to have significantly different clinical outcomes.

Minor Comments

1. Please provide cutoffs for the definitions of BMI categories for perfect clarity.

Reviewer #2 (Remarks to the Author):

This is an interesting large study on rates and risk factors for hospitalization and death in people with covid-19 in the UK.

The results are interesting and useful. I have few comments for your consideration.

- a. Did the analysis of factors associated with mortality and hospitalization (line 191) adjust for testing capacity? positivity rates in the community? and hospital occupancy. These variables affect the probability of being tested and testing positive AND the outcomes, and should be considered analytically.

b. Related to item a, the finding that people living in less deprived areas have less risk of hospitalization and death may be driven by the differential availability of testing and constrained hospital capacity in some areas.

c. The analyses showing temporal changes in hospitalization and death rates are interesting, but do not provide deep insight. A deeper understanding of drivers of the temporal changes is needed. The authors should develop analyses to identify -- to the extent possible -- the major explanatory drivers for this change over time (and how much variance is accounted for, etc...). See this for an example (<https://bmjopen.bmj.com/content/11/8/e047369.info>)

Reviewer #3 (Remarks to the Author):

This paper studies changes in case hospitalisation rate and case fatality rate over time, and investigates whether this can be explained by changes in risk factor distributions over time. It's a nice and interesting piece of work. I have a few suggestions for improvement, mostly around furthering the investigation of the extent to which changes in risk factor distribution over time affect the case hospitalisation rate and case fatality rates.

Comments:

1. The title could do a better job of explaining what the paper is about - i.e. determining how much of the trends/changes over time can be explained by changes in risk factors of those infected over time. Similarly the aim (line 36).
2. The time window for the study (October 2020 - April 2021) is quite narrow. What is the justification for this? Why not conduct the analysis using data from start of pandemic (~Feb 2020) to the most recent data available? Is this a limitation of the data sources used?
3. Line 95: the hospitalisation outcome is a bit unclear - is this within 28 days of positive test? Or where the primary reason for admission is covid-19?
4. Line 125: what is the rationale for including CCG as a random intercept / level? There are a variety of choices that could be made for levels in the hierarchy, e.g. GP practice.
5. Line 131: why was age categorised? Why not use age splines (similar to time)? Same comment for BMI.
6. Line 141: I think the authors mean 'calibration plot'. Hosmer-Lemeshow is a test.
7. Line 142: 'indicated adequate calibration' - that is a Result and should be in Results section.
8. Line 149: first study week is an incomplete week, Thursday-Sunday. I therefore think this is a somewhat confusing choice of referent - e.g. one might expect that this week has unusually low CHR and CFR because of the weekend effect.
9. Line 222: could these findings be expressed as proportion of variation explained? I think that would be very interesting (and easily calculated based on R^2 -like statistics).
10. Fig 1/2: why is age not present in these plots?
11. Fig 3/4 versus S3-S5: for me this is a key comparison. From looking at them side by side it seems apparent that any changes in risk factor distributions over time are having little to no effect on CHR and CFR over time (since the graphs look almost identical). That would be useful to report. So I assume there is actually little change in risk factor distributions over time? This would also be reflected if the authors do an analysis like I suggest in comment 9 above.
12. Finally, for the above analysis the stratification by age group for figures 3/4 etc means we cannot see whether changes in age distribution over time are affecting crude CFR and CHR. That would also be useful to know (as part of comment 9).

Matthew Sperrin

Reviewer #1:

Summary

The manuscript from Beaney et al. is a well written report describing patient-level factors that are associated with hospitalization and death associated with COVID-19 over a 6-month period between October 2020 through April 2021. The major conclusions are relatively consistent with what has been previously published with this large cohort. While the findings of the paper are largely confirmatory re: existing hypotheses or published results, though in a novel data set, if the major comments below are able to be addressed, this would significantly improve the overall impact of the paper and assist in overcoming the limitations highlighted by the authors in the discussion.

Author reply: many thanks for your review and constructive comments. We agree that many of the identified risk factors are consistent with previous studies (such as OpenSAFELY), but the significant trends over time, independent of underlying risk factors, have not been previously documented. Other studies have also been limited by smaller sample sizes, and by the possibility of sampling bias through select populations (eg. patients admitted to hospital), whereas our sample uses the full set of people nationally with confirmed Covid-19 infection. We accept the limitations in our study, which you have highlighted, and have sought to address these more extensively in discussion.

Major Comments

1. The biggest effect modifier for COVID related hospitalization is likely to be the rollout of the vaccination program that overlaps with the observational period, yet this remains relatively unaccounted for in the analysis. The authors describe this appropriately within the discussion as a limitation affecting the interpretation of their results. I think given the known protective benefits from vaccines it's hard to disentangle the periodic time effects vs. introduction of significant new variables that can impact survival/hospitalization. Given there is EHR data on all these patients, is information on vaccination status not available to be added? In the absence of the data maybe there are statistical ways to handle the potential modifying effect. Could analyses consider weighted variable approaches that may help account for likelihood to receive a vaccine – the authors state that the older individuals were the most likely to receive a vaccine such that perhaps age has a differential weight in models according to the time period?

Author reply: we agree that the vaccination programme is a significant explanator for the declines in mortality seen from February 2021 onwards. Unfortunately, we do not have access to individual level vaccination data in our dataset which restricts our ability to look at this factor.

We have considered various approaches for incorporating vaccine data, however each of these approaches would be limited, and would likely bias our results and impact on the interpretability of our findings.

In the absence of individual level data, weekly vaccine coverage data at different levels of granularity (eg national, regional, Sustainability and Transformation Partnership (STP) and Clinical Commissioning Group (CCG)) is available from NHS England. We considered including the weekly vaccine coverage at STP or CCG level within our models (our primary model includes CCG level random intercepts). However, vaccine coverage at a population level is unlikely to be reflective of our population sample. Our study population includes only individuals with a positive Covid-19 test; given that vaccination is expected to reduce infection rates, our population (from 8th December 2020 onwards) are less likely to have received a vaccination than reflected in general population estimates.

A modelling approach to estimate likelihood of vaccination as you suggest, would be interesting, but applying population level estimates to a population with a different underlying risk would require many assumptions and is likely to produce biased inferences. To our knowledge, there is no available data source on vaccine coverage that would be specific to our population (i.e. to people with a positive test).

Use of publicly available vaccine data would also be limited by data availability for our study period. NHS England vaccination data are reported nationally from 27th December 2020 (following the start of the vaccination on 8th December), but STP level data are not available until 17th January 2021, and only for the over 80s vs under 80s, at which point coverage for the 80+ population is at 60%. The availability of data within different age groups varies over time, for example, data on vaccination coverage for the 60-69 age group is not available until 11th March 2021, which is near to our cohort end date (30th April).

We have added into our appendix a set of figures of the changes in total vaccine coverage for the 60-69, 70-79 and 80+ age groups, from the point at which STP level data is available, to give the reader more information on changes in vaccination rates over time. We have also added text to the discussion to provide a greater review of the limitations:

“Declines in the CHR and CFR from January 2021 onwards are likely to be explained at least partially by development of immunity, both through natural infection and by the vaccination programme, which was implemented from 8th December 2020 in England for the highest risk cohorts.²² By February 2021, over 80% of over 80s had been vaccinated in most regions of the

UK, with similar vaccine coverage in the 70-79 year age group by mid-February and in the 60-69 year age group by mid-March (see Figures S8-10 in the supplementary appendix).²³ However, our study population includes people with a positive Covid-19 test, who are more likely to be unvaccinated than the general population; population vaccine coverage is therefore unlikely to be representative of our study population and estimates could not be incorporated robustly into our modelling.”

2. The other major limitation of course is that the authors consider all-cause mortality during the 28-day period of admission. While the authors discuss this limitation, is there no data available on the cause of death? Understanding that post-covid sequelae may result in varying contributions to death, it would still be important data for a reader to understand major cited causes of death, especially among those with significant comorbidity to understand potential differences in risk.

Author reply: thank you for raising this. We have used the 28-day mortality metric in line with UK government and Public Health England reporting. We also have access to data on whether Covid-19 was listed as a cause of death, and have included this in the results. However, we do not have access to the causes of death for people who died without a positive Covid-19 test, which limits inferences on the specific causes of death. We know from UK data that estimates from the Office for National Statistics which use Covid-19 deaths anywhere on the death certificate give a larger total number of Covid-19 deaths than those reported within 28 days of a test.

We have added to the third paragraph of results the proportion of all deaths within 28 days where Covid-19 was listed on the death certificate: “Of these, 49,172 (92.5%) had Covid-19 as a cause of death on the death certificate.” We have also added in the second paragraph of ‘Strengths and limitations’: “Over 90% of deaths within 28 days also had Covid-19 as a cause of death on the death certificate, but we did not have corresponding data for those cases recorded on a death certificate without a positive Covid-19 test. The associations found in our study might be different if using deaths recorded on death certificates, rather than deaths within 28 days of a positive Covid-19 test, particularly if there were changes to death certification practices over time.”

3. Is there a way to do a time-to-event analysis based on the data at hand to understand whether the comorbidities or other factors are associated with greater patient decompensation once admitted? I am envisioning a hazards model where the time-to-event model could be adjusted for the period (month or even week) the patients were admitted with COVID. This may help further disentangle some of the

time effects from evolving secular trends (i.e. changes in treatment; availability of vaccines; etc.) or major circulating variant (emergence of B.1.1.7).

Author reply: we did consider a time to event analysis, however, our chosen outcome was 28-day mortality for consistency with national reporting of Covid-19 mortality in England. We do not think a hazards model would significantly add to the understanding of temporal trends, which are already incorporated in our statistical modelling. As our models do not account for acuity of clinical condition on the date of positive test or arrival at hospital, adopting a time to event analysis is likely to be limited in its validity and is peripheral to the core aims of the study to examine temporal trends in risk of mortality following a positive Covid-19 test. People will have Covid-19 tests done at different stages of illness and shorter time periods may bias estimates in time-to-event analyses to a greater extent than when looking over a longer and fixed time period as used here.

4. More discussion of the clinical relevance of these findings today would be helpful given that we have since moved on from B.1.1.7, experienced/ing delta, and now omicron has swept the world with another variant emerging – all of which seem to have significantly different clinical outcomes.

Author reply: thank you for raising this, and we have included more discussion on the relevance at the end of discussion

Minor Comments

1. Please provide cutoffs for the definitions of BMI categories for perfect clarity.

Author reply: thank you, this was in the appendix and omitted for word count, but has been added to the methods as suggested:

“The emergence of the Delta and Omicron variants have shown the potential of Covid-19 to vary in both transmissibility and pathogenicity over time. In England, December 2021-January 2022 saw the highest case numbers but without the resulting number of hospitalizations and deaths associated with earlier variants and before widespread vaccination.⁴ Despite the emergence of new variants, the findings of our study are relevant in highlighting that risk of mortality was independent of an extensive panel of clinical and demographic factors in the Winter of 2020/21, pointing to the role of wider strain on the health system as an important feature in outcomes in people with Covid-19. While the Omicron variant has contributed to an increase in hospitalisations and emergency department presentations in England and elsewhere, its impact on staff absence has been particularly marked. At the peak of the Omicron wave in early January 2022, almost 50,000 NHS staff were absent due to Covid-19, almost a five-fold increase from the end of November 2021.^{17,26,27} Ensuring health systems possess the resilience

to weather the dual shocks of an increased demand for care and decreased capacity to provide it, without adversely affecting the quality and safety of healthcare, is an ongoing area of concern.”

Reviewer #2:

This is an interesting large study on rates and risk factors for hospitalization and death in people with covid-19 in the UK.

The results are interesting and useful. I have few comments for your consideration.

a. Did the analysis of factors associated with mortality and hospitalization (line 191) adjust for testing capacity? positivity rates in the community? and hospital occupancy. These variables affect the probability of being tested and testing positive AND the outcomes, and should be considered analytically.

Author reply: thank you for these suggestions. Our analysis did not adjust for testing capacity or for positivity rates/hospital occupancy but we did consider incorporating each of these at a CCG or STP level within our models. There is no robust measure for testing capacity or access to testing but there is publicly available data on number of tests and positivity rates. Positivity rates in England peaked on 31st December 2020 at 18.3% and declined to 1.7% by 1st April; this could suggest that those testing positive had a greater severity of illness, if there were limitations in accessing tests, and those with more severe disease were more likely to access one. However, higher positivity rates also reflect higher Covid-19 incidence and it is not possible to distinguish which of these is the case, particularly in the absence of any other marker to indicate disease severity.

We agree that hospital occupancy may impact on a person’s probability of testing and on hospitalisation and mortality. Our study aims to explore outcomes in a population sample (rather than of hospitalised patients as in some other studies) and there is no robust way to determine which hospital a patient in the community should be assigned to. Within the NHS in England, patients may attend any hospital and the size of hospital markets varies considerably by geography, from near monopolies in some, more rural areas, to dilute markets consisting of many hospitals with overlapping, ill-defined catchment areas. Changes in people’s behaviours in response to hospital occupancy are likely to be significantly influenced by socio-demographic factors, such as willingness to travel, opportunity to travel and knowledge of nearby hospital

capacity. As such, it is not possible to reliably model an individual's nearby bed availability in a manner which reflects their variable capacity to fully exploit any available capacity. We have added a discussion of these to paragraph 3 of 'Strengths and limitations':

“Geographical and time-varying system factors, such as proximity to a hospital and hospital capacity are likely to impact on a person's health-seeking behaviour. Our study included people living in the community and given patients in England may attend any hospital, and the size of hospital markets vary considerably across the country, we could not reliably model the impact of nearby hospital bed availability at an individual level.”

b. Related to item a, the finding that people living in less deprived areas have less risk of hospitalization and death may be driven by the differential availability of testing and constrained hospital capacity in some areas.

Author reply: we agree this could be the case and have discussed this in more detail in Discussion:

“Access to testing may also impact the probability of having a positive test. Positivity rates in England peaked on 31st December 2020 at 18.3% and fell to 1.7% by 1st April 2021, but the extent to which this reflects increased incidence or a lack of test availability is uncertain. It is possible that if testing were limited during the peak in cases in December 2020-January 2021, those with more symptomatic disease may have been more likely to receive a test, compared to those who were asymptomatic or with mild symptoms. This may in turn lead to an apparent increase in risk of mortality due to changes in the severity of illness of those testing positive, rather than the severity of disease within the community as a whole. Furthermore, access to testing may be driven by sociodemographic factors, and the finding of lower hospitalisation and mortality risk in less deprived areas could reflect better availability of testing.”

c. The analyses showing temporal changes in hospitalization and death rates are interesting, but do not provide deep insight. A deeper understanding of drivers of the temporal changes is needed. The authors should develop analyses to identify -- to the extent possible -- the major explanatory drivers for this change over time (and how much variance is accounted for, etc...). See this for an example (<https://bmjopen.bmj.com/content/11/8/e047369.info>)

Author reply: we thank the review for providing this suggestion and the helpful reference, which we have added in support of similar findings from previous literature in paragraph 3 of discussion “with similar findings from a US cohort between March and September 2020”. Our paper aims to identify whether individual patient underlying risk is related to mortality trends over time. As mentioned above, given our population sample, rather than use of exclusively hospitalised patients, we cannot robustly explore the impact of hospital occupancy on outcomes at the individual level. As we raised in our comments to reviewer 1, we also believe that by

applying population level estimates to an individual patient analysis would introduce bias. Unlike in the paper above, our aim was not to predict the proportion of all variance explained by each risk factor at a population level, but to identify the associations of each risk factor with outcomes at an individual level and identify whether risk varied over time.

Reviewer #3:

This paper studies changes in case hospitalisation rate and case fatality rate over time, and investigates whether this can be explained by changes in risk factor distributions over time. It's a nice and interesting piece of work. I have a few suggestions for improvement, mostly around furthering the investigation of the extent to which changes in risk factor distribution over time affect the case hospitalisation rate and case fatality rates.

Author reply: many thanks for your feedback and constructive comments.

Comments:

1. The title could do a better job of explaining what the paper is about - i.e. determining how much of the trends/changes over time can be explained by changes in risk factors of those infected over time. Similarly the aim (line 36).

Author reply: we have kept the title short, given the advice for 15 words or under, but are happy to adapt it if the editors prefer. Similarly with the abstract, we are happy to adapt as per journal preferences.

2. The time window for the study (October 2020 - April 2021) is quite narrow. What is the justification for this? Why not conduct the analysis using data from start of pandemic (~Feb 2020) to the most recent data available? Is this a limitation of the data sources used?

Author reply: This was a limitation of the data sources. The work was carried out as part of a wider evaluation of an NHS remote monitoring programme, which started in November 2020, and so our data collection was specific for this period.

3. Line 95: the hospitalisation outcome is a bit unclear - is this within 28 days of positive test? Or where the primary reason for admission is covid-19?

Author reply: thanks for highlighting this, we agree the text was unclear and have amended to make clear the 28 days refers to each outcome

4. Line 125: what is the rationale for including CCG as a random intercept / level? There are a variety of choices that could be made for levels in the hierarchy, e.g. GP practice.

Author reply: There are a variety of choices that could be made here, and we chose to use a healthcare organisational hierarchy, so STP, CCG or GP practice. There are 42 STPs in England, 106 CCGs and >6000 GP practices and we chose CCGs given the greater granularity to STPs and due to likely data sparseness for some GP practices.

5. Line 131: why was age categorised? Why not use age splines (similar to time)? Same comment for BMI.

Author reply: we wanted to keep age and BMI interpretable, particularly given we were already plotting time as a spline on the x-axis. Both the BMI categories and the age categories have a clinically meaningful interpretation, whereas the coefficients for the splines themselves have very little interpretability.

6. Line 141: I think the authors mean 'calibration plot'. Hosmer-Lemeshow is a test.

Author reply: thank you, we have corrected this in the manuscript and the appendix.

7. Line 142: 'indicated adequate calibration' - that is a Result and should be in Results section.

Author reply: we have moved this to results as suggested.

8. Line 149: first study week is an incomplete week, Thursday-Sunday. I therefore think this is a somewhat confusing choice of referent - e.g. one might expect that this week has unusually low CHR and CFR because of the weekend effect.

Author reply: we agree and have changed the relative risk estimates to the 'first full study week' (5th-11th October) throughout. This changes the plots showing relative risk, but all other figures (including the estimates presented in the tables) are based on the absolute risk which does not change.

9. Line 222: could these findings be expressed as proportion of variation explained? I think that would be very interesting (and easily calculated based on R²-like statistics).

Author reply: With linear regression, the R² statistics can be interpreted as the proportion of variance explained. Although pseudo-R² statistics are often used in logistic regression, these do not have a direct interpretation as proportion of variance explained, as in linear regression. As discussed in Giselmar et al (2016) - doi:10.1177/0049124116638107 , its use is most suited for comparison of models, rather than in the metric itself. Our aim was to present the measures of association of theoretically defined sets of confounders, rather than prediction and so the R² value of the model has limited value in interpretation.

10. Fig 1/2: why is age not present in these plots?

Author reply: our models included age category with an interaction with the (four) time splines. Because of this, the interpretation of the coefficient values have little intrinsic meaning, and so we have presented these in the figures instead (over time, and stratified by age category). We believe that including the estimates in the plots/tables could confuse the reader and lead to erroneous interpretation of the age category estimates.

11. Fig 3/4 versus S3-S5: for me this is a key comparison. From looking at them side by side it seems apparent that any changes in risk factor distributions over time are having little to no effect on CHR and CFR over time (since the graphs look almost identical). That would be useful to report. So I assume there is actually little change in risk factor distributions over time? This would also be reflected if the authors do an analysis like I suggest in comment 9 above.

Author reply: thanks for highlighting these points, it was implied but not clearly stated in the results. We have added to the end of results:

“After adjustment, the trends in absolute mortality and hospitalisation risk in each age group were similar to those in the unadjusted analyses (Figure 4 and Figure S4) indicating that the distributions of risk factors of those infected within age groups did not change significantly over time.”

12. Finally, for the above analysis the stratification by age group for figures 3/4 etc means we cannot see whether changes in age distribution over time are affecting crude CFR and CHR. That would also be useful to know (as part of comment 9).

Author reply: thank you for suggesting this. Analyses of the whole population were part of our preliminary work but not added to the paper in the interests of keeping it focussed. However, we agree that this is a natural question to follow and have added into the appendix. We have added the distribution of age of those infected over time (Table S1), along with plots of the CFR and CHR for the whole population (i.e. including all age groups) for unadjusted (time only, Figure S3) and fully adjusted (as with the existing analyses, Figure S5) models. As shown, the relative risk increases at a whole population level, but is attenuated somewhat following adjustment. The ‘bimodal’ distribution of the CFR and CHR in unadjusted models are also a feature of changing patterns of age groups infected, as these become unimodal following adjustment. We have added text into the results to present the findings, and referenced the figures in the appendix in the first paragraph of ‘CHR and CFR over time’ within results:

“The predicted CHR and CFR from the fully adjusted models are plotted for the whole population (Supplementary Appendix Figure S5) and by age category in Figure 3, showing that a significant time-varying relationship remained after adjustment.”

We have also commented on this in the first paragraph of ‘Variation in CHR and CFR over time’ in Discussion:

“Across the whole study population, CHR and CFR varied over time from 1st October 2020 to 30th April 2021. This was partially explained by the changing age distributions of those infected, but significant variation remained after adjustment. Within age groups, absolute differences in the CHR and CFR over time were greatest in older age groups, reflecting higher baseline risk, but the relative risk varied significantly across all groups.”

REVIEWERS' COMMENTS

Reviewer #1 (Remarks to the Author):

The authors have adequately addressed my prior comments. Well done on an interesting piece of work.

Reviewer #2 (Remarks to the Author):

The authors did not undertake additional analyses in response to my comments, but added several sentences to the Discussion section. their inability to add more analyses is likely constrained by availability of data (positivity rates, etc..). I think their responses are reasonably satisfactory. Defer to your ultimate assessment. Thanks again for inviting me to review this, and happy to help in any way.

Reviewer #3 (Remarks to the Author):

The authors have done a good job in addressing my comments - I have nothing further to add.